# The Role of Diffusion-Weighted Imaging Based on Maximum-Intensity Projection in Young Patients with Marked Background Parenchymal Enhancement on Contrast-Enhanced Breast MRI

**DOI:** 10.3390/life13081744

**Published:** 2023-08-14

**Authors:** Ga-Eun Park, Bong-Joo Kang, Sung-hun Kim, Na-Young Jung

**Affiliations:** 1Department of Radiology, Seoul Saint Mary’s Hospital, College of Medicine, The Catholic University of Korea, Seoul 06591, Republic of Korea; hoonhoony@naver.com (G.-E.P.); lionmain@catholic.ac.kr (B.-J.K.); rad-ksh@catholic.ac.kr (S.-h.K.); 2Department of Radiology, Uijeongbu Eulji Medical Center, College of Medicine, Eulji University, Uijeongbu 11759, Republic of Korea

**Keywords:** breast cancer, MRI, diffusion-weighted

## Abstract

Diffusion-weighted imaging (DWI) utilizing maximum-intensity projection (MIP) was suggested as a cost-effective alternative tool without the risk of gadolinium-based contrast agents. The purpose of this study was to investigate whether DWI MIPs played a supportive role in young (≤60) patients with marked background parenchymal enhancement (BPE) on contrast-enhanced MRI (CE-MRI). The research included 1303 patients with varying degrees of BPE, and correlations between BPE on CE-MRI, the background diffusion signal (BDS) on DWI, and clinical parameters were analyzed. Lesion detection scores were compared between CE-MRI and DWI, with DWI showing higher scores. Among the 186 lesions in 181 patients with marked BPE on CE-MRI, the main lesion on MIPs of CE-MRI was partially or completely seen in 88.7% of cases, while it was not seen in 11.3% of cases. On the other hand, the main lesion on MIPs of DWI was seen in 91.4% of cases, with only 8.6% of cases showing no visibility. DWI achieved higher scores for lesion detection compared to CE-MRI. The presence of a marked BDS was significantly associated with a lower likelihood of a higher DWI score (*p* < 0.001), and non-mass lesions were associated with a decreased likelihood of a higher DWI score compared with mass lesions (*p* = 0.196). In conclusion, the inclusion of MIPs of DWI in the preoperative evaluation of breast cancer patients, particularly young women with marked BPE, proved highly beneficial in improving the overall diagnostic process.

## 1. Introduction

Contrast-enhanced magnetic resonance imaging (CE-MRI) of the breast has the highest sensitivity for breast cancer detection among several imaging modalities [1,2]. Despite its highest degree of sensitivity, breast MRI does not result in false-negative cases in comparison to other imaging modalities. False-negative cases may be attributed to perceptive errors in the absence of radiological detection at the time of screening; interpretation errors, where the cases are recognized but mistaken for benign lesions; and various technical errors [3]. A recent study reported three main causes of undetected breast malignancy in CE-MRI: (1) non-enhancing histologic features; (2) location; and (3) significant background parenchymal enhancement (BPE) [4]. BPE significantly affects breast MRI interpretation and is a valuable imaging marker for assessing breast cancer risk [5]. BPE is widely recognized for increasing the recall, false-positive, and false-negative rates in breast MR readings. In particular, substantial BPE may prevent the clear demarcation of lesions from the breast parenchyma [6].

Several earlier studies have demonstrated that the combination of a diffusion-weighted imaging (DWI) protocol with CE-MRI leads to higher specificity compared to CE-MRI alone [7,8,9,10]. Furthermore, recent research has indicated its potential for breast cancer detection and the characterization of breast lesions [11,12].

The abbreviated protocol (AP) for breast MRI, utilizing a single pre-contrast and a single post-contrast acquisition along with maximum-intensity projection (MIP) images, has become increasingly popular. The advantages include a shorter acquisition and reading time, lower cost, and diagnostic accuracy comparable to the full protocol [13,14]. So, DWI MIPs could be proposed as a cost-effective alternative tool, eliminating the risk associated with gadolinium-based contrast agents in this study.

Therefore, the objective of this study was to assess the effectiveness of DWI utilizing MIPs in distinguishing lesions compared with the conventional protocol of using MIPs of CE-MRI for patients with preoperative breast cancer, with a specific focus on young women presenting marked BPE.

## 2. Materials and Methods

### 2.1. Study Population

This study was approved by our Institutional Review Board, and informed consent was waived due to its retrospective nature. The study period spanned from 1 July 2020 to 30 September 2022, and a total of 4199 MRI scans were included. Among these, 1712 scans were performed preoperatively. A total of 409 cases were excluded such as those associated with neoadjuvant chemotherapy (NAC) (309 cases), post-excision scans (33 cases), and specific criteria such as old age (>60 years) (51 cases), inflammatory cancers (9 cases), implants (5 cases), and absence of surgical confirmation (2 cases).

### 2.2. MRI Protocol

Breast MRI was performed using 3T MR machines (Verio and Vida, Siemens Healthcare, Erlangen, Germany). Breast MRI scans were conducted in the prone position using a specialized breast surface coil. The enrolled patients underwent the following MRI sequences for the Verio system: (1) Axial T2-weighted imaging with a turbo spin-echo technique, using a TR/TE of 4530/93, a flip angle of 80, 34 slices, a 320 mm field of view, a matrix size of 576 × 403, 1 excitation, a 4 mm slice thickness, and an acquisition time of 2 min 28 s. (2) Axial DWI with a readout-segmented echoplanar image, employing b values of 0 and 1000 s/mm^2^, a TR/TE of 5200/53 ms, a field of view of 340 × 205 mm^2^, a matrix size of 192 × 116, a 4 mm slice thickness, and an acquisition time of 2 min 31 s with 5 readout segments. The apparent diffusion coefficient (ADC) maps were automatically calculated using software. (3) Pre- and post-contrast axial T1-weighted 3D volumetric interpolated brain examination (VIBE) sequences with a TR/TE of 2.7/0.8, a flip angle of 10, and a 1.2 mm slice thickness. The images were acquired before and at 10, 70, 130, 190, 250, and 310 s after the injection of gadolinium DTPA (0.1 mmol/kg of Gadovist; Bayer Schering Pharma, Berlin, Germany). For the Vida system, the MRI sequences were as follows: (1) Axial T2-weighted imaging with a turbo spin-echo DIXON sequence, using a TR/TE of 5000/96 ms, a flip angle of 120, 50 slices, a 320 mm field of view, a matrix size of 448 × 314, a 3 mm slice thickness, and an acquisition time of 3 min 23 s. (2) Axial DWI with readout-segmented long variable echo trains, employing b values of 0 and 1000 s/mm^2^, a TR/TE of 4720/60 ms, a field of view of 350 × 210 mm^2^, a matrix size of 256 × 154, a 3 mm slice thickness, and an acquisition time of 3 min 29 s with 9 readout segments. The ADC maps were automatically calculated using software. (3) Pre- and post-contrast axial T1-weighted 3D fast low-angle-shot (FLASH) sequences with a TR/TE of 4.7/2.27 ms, a flip angle of 10, and a 1 mm slice thickness. The images were acquired before and at 10, 93, 176, 259, 342, and 425 s after the injection of gadolinium DTPA.

Digital Imaging and Communications in Medicine (DICOM) files from DCE-MRI and DWI were transferred to a computer software program (Syngovia; Simens healthcare, Erlangen, Germany) in order to generate MIP images using high b-value DWI and first postcontrast subtracted images.

### 2.3. Image Analysis

Among the 1303 enrolled breast MRI results, breast parenchymal enhancement (BPE) and background diffusion signal (BDS) were assessed by one of three radiologists with 5–20 years of experience in breast imaging. The degrees of BPE and BDS were categorized as minimal, mild, moderate, or marked according to the American College of Radiology Breast Imaging and Reporting Data System (BI-RADS) [4]. In cases where BPE or BDS exhibited asymmetry between bilateral scans, the breast with the highest BPE or BDS was utilized for categorization purposes.

Two breast radiologists with experiences of 18 years and 20 years reviewed the two sets of images and arrived at a consensus based on MIPs of DWI and CE-MRI. The readers were blinded to the women’s clinical histories and other imaging sets. The images were transferred to numbered folders containing anonymized image data on a Picture Archiving and Communication System (PACS) and read according to the following standardized protocol.

First, the readers reviewed the MIP of DWI to identify significant lesions: (1) definitively seen group, based on the consensus of two readers’ findings of suspicious lesions; partial or retrospectively seen group, based on findings of suspicious lesion by one out of two readers; (2) unseen or undetected group, if neither reader found a true lesion. The readers characterized (mass/non-mass) and scored (1 to 10) the detected lesions on DWI MIPs.

Second, the readers reviewed the MIPs of CE-MRI obtained under early enhancement similar to DWI MIPs. The MIPs did not allow a full assessment of lesion morphology; thus, we used a scoring system rather than BI-RADS. Multiple breast lesions were divided into primary breast cancers as main lesions and additional suspicious lesions as daughter lesions.

Image analysis was conducted using a 1–10 scoring system. The scores of CE-MRI and DWI were used to determine lesion visibility and characteristics. The score ranged from 1 to 10, with corresponding descriptions as Table 1.

The image gold standard for true lesions was established using conventional whole CE-MRI, mammography, and ultrasonography. The other radiologist, with five years of experience, analyzed sets of images and compared the results with the image gold standard.

Ductal carcinomas in situ (DCISs) and invasive cancers were counted as positive results. All other results of biopsy or excision analysis, including high-risk lesions such as atypical ductal hyperplasia, lobular carcinoma in situ, or papilloma, were considered as negative results.

We analyzed the correlation between BPE depending on MIP images of CE-MRI, BDS on MIP images of DWI, and clinical parameters such as age. Additionally, we analyzed the scores of primary breast cancer and examined additional suspicious lesions on both CE-MRI and DWI. Finally, the features of malignant breast lesions were also analyzed.

### 2.4. Histopathology Review

The biopsy or surgical specimen pathology reports were carefully examined to determine various tumor characteristics, such as size, depth, histologic type, grade, presence of lymph node metastasis, and immunohistochemical (IHC) subtypes. The IHC factors evaluated included estrogen receptor (ER), progesterone receptor (PR), human epidermal growth factor receptor 2 (HER2), Ki-67, epidermal growth factor receptor (EGFR), and CK5/6. IHC staining for ER, PR, HER2, Ki-67, and EGFR was conducted using specific primary antibodies on an automated Ventana BenchMark XT Slide Stainer (Ventana, Tucson, AZ, USA). The staining for CK5/6 was performed on the Dako Omnis (Dako, Carpinteria, CA, USA). For ER and PR positivity, a cut-off value of ≥1% was used. HER2 expression intensity was semiquantitatively scored as 0, 1, 2, or 3, with a score of 3 indicating HER2 positivity, while scores of 0 or 1 indicated HER2 negativity. HER2 status for tumors with a score of 2 was determined using gene amplification [15]. Positive Ki-67 expression was defined as Ki-67 positivity in ≥14% of cancer cell nuclei. EGFR and CK5/6 positivity were defined with a cut-off value of ≥1%. Based on the 2013 St. Gallen International Breast Cancer Conference recommendations, the IHC subtypes were classified as follows [16]: (1) Luminal A (ER or PR+, HER2-, and Ki-67low), (2) Luminal B (ER or PR+, HER2+, and/or Ki-67high), (3) HER2+ (ER-, PR-, and HER2+), (4) triple-negative basal-like (ER-, PR-, HER2-, EGFR, or CK5/6+), and (5) triple-negative non-basal-like (ER-, PR-, HER2-, EGFR-, CK5/6-).

### 2.5. Statistical Analysis

Summary statistics are presented as number (percentage) of categorical variables and as means (standard deviation (SD)) and medians (inter-quartile range (IQR)) in the case of continuous variables. Groups were compared using the chi square test or Fisher’s exact test and Wilcoxon rank-sum tests for categorical and continuous variables, respectively. Comparison of CE-MRI and DWI was performed using the generalized estimating equation (GEE) or the cluster Wilcoxon rank-sum test, considering the same subjects as clusters, *p* value (CE-MRI vs. DWI in unseen category), and *p* value (CE-MRI vs. DWI in seen category). To investigate factors associated with higher score in DWI group than in CE-MRI, univariate and multivariable logistic regression analyses were performed. Variables were included in the multivariable model if their univariate significance was <0.05. All statistical analyses were performed using SAS version 9.4 (SAS Institute, Cary, NC, USA), with two-sided *p*-values < 0.05 considered statistically significant.

## 3. Results

The results of the study are summarized in Figure 1, presenting the flow diagram of the study population selection. A total of 1303 scans exhibited varying degrees of BPE, categorized as minimal (n = 422), mild (n = 410), moderate (n = 290), and marked (n = 181) (Figure 2). Within the marked BPE group, bilateral cases were analyzed separately (n = 3). Cases involving ipsilateral multiple masses with different pathologies were also analyzed separately (n = 2). In total, this study assessed 186 lesions in 181 patients with marked BPE on CE-MRI.

The results of the study are presented in Table 2, providing important clinical, imaging, and pathologic characteristics of a total of 186 lesions included in the analysis. The mean age of the patients was 44.6 years (SD = 5.8), with a median of 46.0 years (IQR: 42.0–48.0). MRI revealed that the mean size of the lesions was 28.1 mm (SD = 19.3), with a median size of 21.5 mm (IQR: 15.0–36.0). Among the lesions, 66.1% were categorized as masses, 27.4% as non-masses, and 6.5% as both masses and non-masses. The mean ADC value was 0.9 × 10^−3^ mm^2^/s (SD = 0.2), with a median of 0.9 × 10^−3^ mm^2^/s (IQR: 0.8–1.0). The average CE-MRI score was 5.5 (SD = 2.0), with a median score of 6.0 (IQR: 4.0–7.0). The mean score in DWI was 6.2 (SD = 2.3), with a median score of 7.0 (IQR: 4.0–8.0). The combined score, considering both CE-MRI and DWI, had a mean of 6.6 (SD = 2.1) and a median of 7.0 (IQR: 5.0–8.0). Regarding pathologic characteristics, the majority of lesions were invasive ductal carcinoma (IDC) (67.7%), followed by ductal carcinoma in situ (DCIS) (23.7%). Immunohistochemical subtypes revealed 50.0% classified as Luminal A, 37.1% as Luminal B, 4.3% as HER2-positive, and 8.6% as triple-negative.

Table 3 presents several clinically important findings regarding the detectability of the main lesion on MIPs of CE-MRI and MIPs of DWI. Among the total of 186 cases analyzed, the main lesion on MIPs of CE-MRI was partially or definitely seen in 165 cases (88.7%), while it was not seen in 21 cases (11.3%). However, the main lesion on MIPs of DWI was partially or definitely seen in 170 cases (91.4%), with only 16 cases (8.6%) showing no visibility (Figure 3). Statistical analysis revealed a significantly different detection of the main lesion on MIPs of CE-MRI based on size (*p* = 0.002), where the mean size of visible lesions was 29.4 mm compared with 18.4 mm for invisible lesions. However, the detectability based on size was not significantly different for the main lesion on MIPs of DWI (*p* = 0.157). No statistically significant differences were found in the detectability of the main lesion on MIPs of CE and MIPs of DWI based on the mass/non-mass categorization (*p* = 0.092 and *p* = 0.146, respectively). The ADC values showed no significant difference in detectability of the main lesion on MIPs of CE-MRI (*p* = 0.235), while a significant difference was observed for the main lesion on MIPs of DWI (*p* = 0.017). The combined scores of CE-MRI and DWI showed significant differences in detectability for both MIPs of CE-MRI and MIPs of DWI (*p* < 0.001). Notably, the detectability of the main lesion on MIPs of CE-MRI and DWI was associated with pathology, with statistically significant differences observed for both (*p* = 0.012 and *p* = 0.216, respectively). In summary, the study findings highlight the influence of size, ADC values, combined scores, and pathology on the detectability of the main lesion on MIPs of CE-MRI and DWI, providing valuable insights for the clinical assessment and interpretation of breast MRI results.

Univariable and multivariable logistic regression analyses were conducted to identify clinically important findings associated with a higher DWI score than CE-MRI score (Table 4). Among the variables examined, two variables showed significant association in the multivariable analysis. First, the presence of a marked BDS was significantly associated with a lower likelihood of a higher DWI score (odds ratio [OR] = 0.18, 95% confidence interval [CI] = 0.08–0.38, *p* < 0.001) (Figure 4 and Figure 5). Second, the presence of a non-mass lesion was associated with a decreased likelihood of a higher DWI score compared with mass lesions (OR = 0.61, 95% CI = 0.28–1.29, *p* = 0.196) (Figure 6). Other variables, including size, ADC value, pathology type, tumor grade, lymph node status, hormone receptor status (ER and PR), HER2 status, KI-67 group, and immunophenotype, did not show a statistically significant association with higher DWI scores in the multivariable analysis (*p* > 0.05).

## 4. Discussion

This study showed that the MIP of DWI significantly improved the detection of true suspicious lesions on MR imaging of young women with marked BPE.

Since its introduction in 1986, contrast-enhanced breast MRI has become the most sensitive method for detecting invasive breast cancer [17,18,19]. After the intravenous administration of a gadolinium-based contrast agent, BPE can lead to the enhancement of normal breast fibroglandular tissue. The extent of BPE can differ among women and even within the same individual, and it is believed to be associated with changes in the vascular supply and permeability of breast tissue, which are influenced by hormonal status [5]. However, BPE of normal breast parenchyma is a well-known and major clinical concern, significantly limiting breast tumor detection using CE-MRI. Telegrafo et al. and Demartini et al. reported that moderate and marked BPE reduces the sensitivity of CE-MRI imaging when compared with minimal and mild BPE [20,21]. Many earlier studies have shown that combining the DWI protocol results in a higher specificity compared with CE-MRI alone, by reducing false positives [7,8,9,10].

The evolving approach of using an AP for screening breast MRI offers several advantages, including shorter acquisition and interpretation times, reduced cost, and comparable diagnostic accuracy to the full protocol [13,14]. In a previous study, the AP used one pre-contrast and one post-contrast acquisition with MIP images, completing the MRI acquisition in just 3 min and the interpretation time in less than 30 s. Remarkably, the diagnostic performance was on par with the full protocol [14]. Additionally, recent apprehensions regarding the deposition of gadolinium-based contrast agents in neuronal tissues must not be dismissed [22]. Conversely, DWI is a valuable unenhanced technique that offers microstructural insights at the cellular level, enabling the detection of changes in tissue water related to modifications in tissues and intracellular structures [11]. Recent studies have demonstrated its potential for detecting and characterizing breast lesions, with technical advances enhancing its quality. In uncertain cases, the ADC on DWI can be utilized to reduce the need for biopsies [11,12,23,24]. A significant advantage is its high sensitivity for detecting breast cancer without the need for contrast material injection, as shown in a recent meta-analysis with an overall sensitivity of 84% and specificity of 79% [25]. Our study also detected a significant difference in the main lesion on MIPs of DWI based on ADC values (*p* = 0.017).

Kang et al. suggested that DWI MIPs could be a cost-effective alternative to the AP, leading to shorter acquisition and interpretation times, while avoiding the risk of gadolinium-based contrast agents. In their study, the AP’s diagnostic performance, employing T1WI and rs-EPI DWI, closely resembled that of conventional CE-MRI, showcasing sensitivities ranging from 80.0% to 90.0%, specificities from 93.4% to 95.1%, PPV3s from 28.1% to 32.0%, and NPVs from 99.4% to 99.7%. The false-positive rates were minimal, ranging from 4.7% to 6.4% [26]. However, their study population differed from ours, as it involved postoperative breast MRI, which is generally easier due to treatment-related changes such as decreased lesion numbers after surgery and reduced BPE following radiation or anti-hormonal therapy [19].

Our study focused on preoperative CE-MRI, and the frequencies of various BPE categories were as follows: minimal (n = 422, 32.4%), mild (n = 410, 31.5%), moderate (n = 290, 22.3%), and marked (n = 181, 13.9%), with marked BPE showing the least prevalence. We specifically targeted young patients with breast cancer (≤60 years old) with marked BPE. BPE is a valuable imaging marker for breast cancer risk assessment and can impact the interpretation of breast MRI [27,28]. And BPE is a known risk factor for breast cancer [5,29,30]. In our institution, the frequency of breast cancer in individuals over the age of 60 is significantly lower. The elderly population (>60) represents only a small proportion, and instances of marked BPE in this age group are exceedingly rare. Therefore, for this study, we defined the elderly age group as >60 and excluded it accordingly.

The MIP was central to our study, and its value has been demonstrated. The MIP images facilitated easy and rapid assessment and comparison. However, MIP images have limitations in evaluating shape, margin, and internal enhancement compared with the entire conventional CE-MRI images. Therefore, instead of using the BI-RADS, we employed a 1–10 scoring system, focusing on detectability. In our study, among 181 patients with marked BPE on the MIP of CE-MRI, the distribution of the BDS was as follows: minimal (n = 18, 9.7%), mild (n = 47, 25.3%), moderate (n = 58, 31.2%), and marked (n = 63, 33.9%), with the majority being unmarked (123, 66.1%) compared with marked (63, 33.9%). The observed results indicate a lack of correlation between BPE on CE-MRI and the BDS on DWI. While mammographic density and BPE are well-established risk factors for breast cancer, no significant correlation was found between them [5]. Consequently, we propose that mammographic density, BPE, and BDS are not correlated factors. Based on these findings, we confirmed that DWI is particularly helpful in breast cancer detection, especially in patients with marked BPE on CE-MRI but an unmarked BDS on DWI. Accordingly, the univariable and multivariable logistic regression analyses were conducted to identify clinically important findings associated with a higher score in DWI than in CE-MRI. Among the variables examined, two variables showed a significant association in the multivariable analysis. First, the presence of a marked BDS was significantly associated with a lower likelihood of a higher DWI. Second, the presence of a non-mass lesion was associated with a decreased likelihood of a higher DWI score compared with a mass lesion.

There were several limitations to our present study. First, we included a limited number of patients and adopted a retrospective study design. However, we did include a consecutive group of uniform patients who underwent preoperative breast MRI using a 3T MR scanner during the study period and definitive surgery. Second, we evaluated only the preoperative breast MRI, which may induce selection bias. This may affect image evaluations by radiologists. Third, subjective interpretations of MR imaging may also affect the image evaluation by radiologists.

Our primary focus in this study was to assess the detectability of breast cancer in young patients with marked BPE. We did not find any correlation between BPE on CE-MRI and a BDS on DWI. However, in cases where CE-MRI showed marked BPE but DWI did not show a marked BDS, additional analysis of DWI proved to be extremely helpful in breast cancer detection. Furthermore, we found that utilizing MIPs of DWI served as an effective tool for detecting breast malignancy.

## 5. Conclusions

In conclusion, the inclusion of MIPs of DWI in the preoperative evaluation of breast cancer patients, particularly young women with marked BPE, can be highly beneficial in improving the overall diagnostic process.

## Figures and Tables

**Figure 1 life-13-01744-f001:**
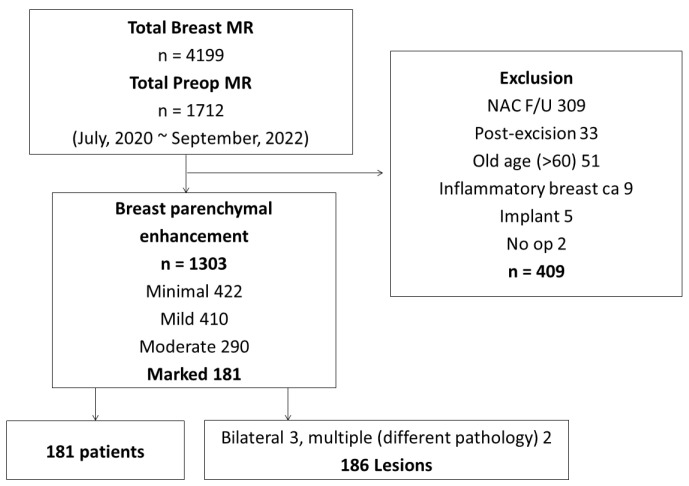
Flow diagram of study population selection.

**Figure 2 life-13-01744-f002:**
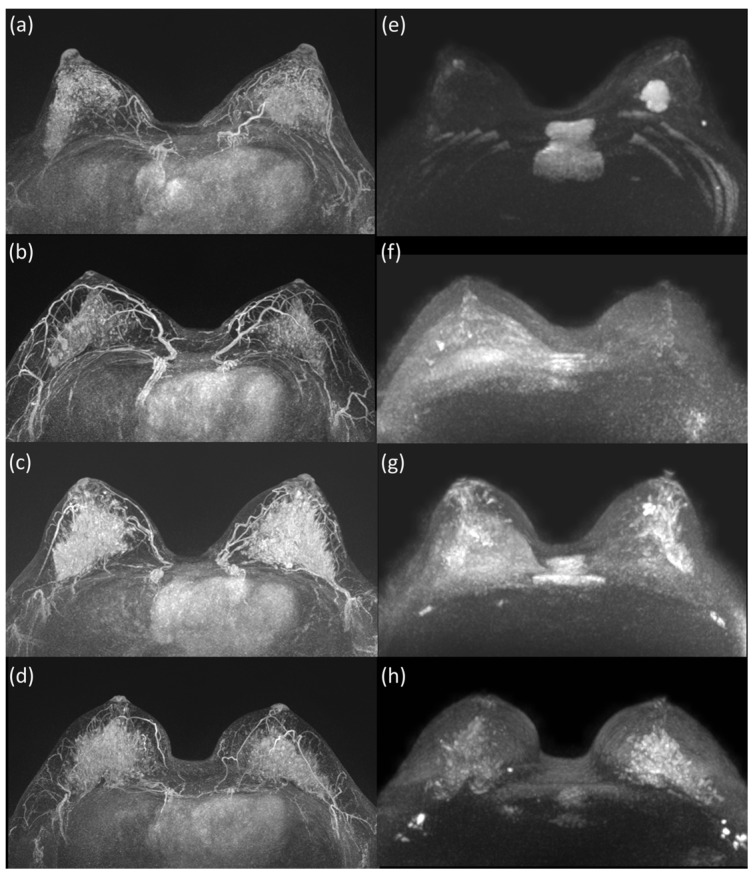
(**a**–**d**) Examples of marked breast parenchymal enhancement (BPE) on MIP of CE-MRI with (**e**) minimal, (**f**) mild, (**g**) moderate, and (**h**) marked background diffusion signal (BDS) on MIP of DWI.

**Figure 3 life-13-01744-f003:**
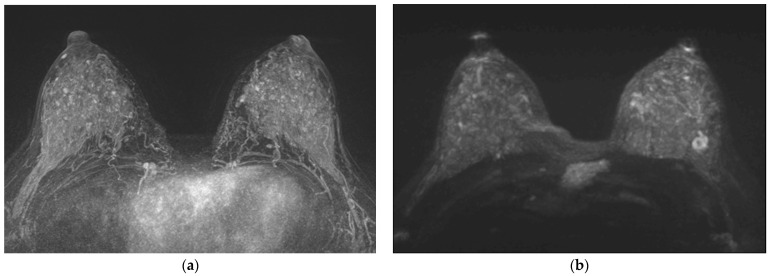
Preoperative breast MRI of 39-year-old woman with invasive ductal carcinoma. (**a**) MIP of CE-MRI showed marked BPE and unseen suspicious breast lesion, (**b**) MIP of DWI showed mild BDS and a 1.3 cm mass in the 3 o’clock position of left breast. ADC value on ADC map was 0.73. In the scoring system, 1 on CE-MRI, 9 on DWI and combined. Triple-negative breast cancer was confirmed.

**Figure 4 life-13-01744-f004:**
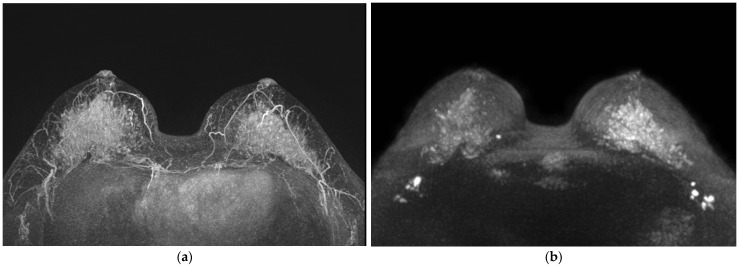
Preoperative breast MRIs of 43-year-old woman with breast malignant lesions. The presence of a marked BDS was significantly associated with a lower likelihood of a higher DWI score. (**a**) MIP of CE-MRI showed marked BPE and unseen suspicious breast lesion (score 1), (**b**) MIP of DWI showed marked BPE and unseen suspicious breast lesion (score 1). About 1.2 cm mucinous cancer was confirmed at the 5 o’clock position in left breast.

**Figure 5 life-13-01744-f005:**
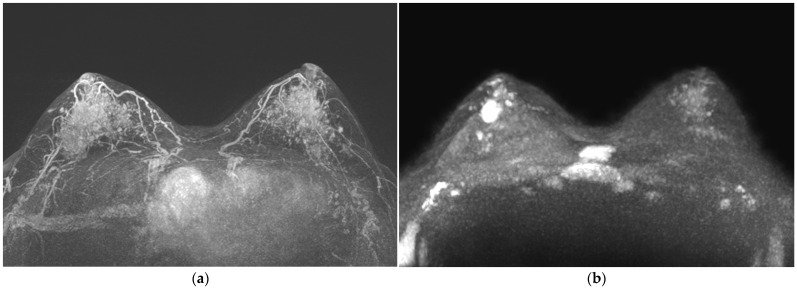
Preoperative breast MRIs of 49-year-old woman with breast malignant lesions. The presence of a marked BDS was significantly associated with a lower likelihood of a higher DWI score. (**a**) MIP of CE-MRI showed marked BPE and partially seen suspicious breast lesion (score 4), (**b**) MIP of DWI showed mild BDS and definitive suspicious breast lesion with false-positive lesions (score 7). About 1.5 cm-sized invasive ductal cancer was confirmed in the central portion in right breast.

**Figure 6 life-13-01744-f006:**
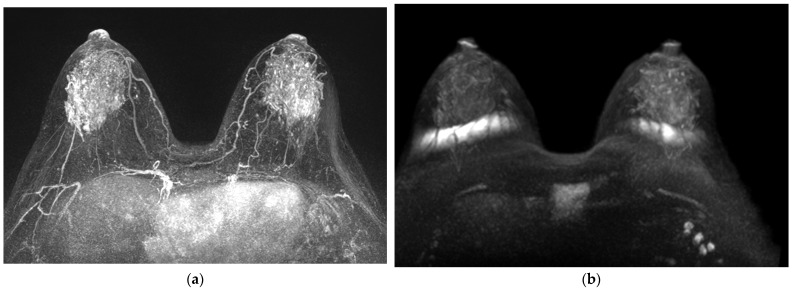
Preoperative breast MRI of a 52-year-old woman with ductal carcinoma in situ. The presence of a non-mass lesion was associated with a decreased likelihood of higher DWI score compared with the presence of mass lesions. (**a**) MIP of CE-MRI showed marked BPE and partially seen segmental non-mass enhancement lesion (score 4), (**b**) MIP of DWI showed mild BDS and unseen suspicious breast lesion (score 1). DCIS measuring about 4.5 cm was confirmed in the outer portion of the right breast.

**Table 1 life-13-01744-t001:** Definition of group and score.

Group	Score	Definition
Unseen	1	absolutely not seen
	2	very subtle visibility
Partially or retrospectively seen	3	partial visibility where the lesion was only visible when its location was known (less than 50% visibility)
	4	partial visibility where the lesion was visible when its location was known (more than 50% visibility)
	5	signified complete visualization of the entire lesion when its location was known, retrospectively
Definitely seen	6	visualization of the lesion similar to moderate BPE on CE-MRI, albeit with numerous false-positive lesions
	7	complete confirmation of the lesion similar to moderate BPE on CE-MRI, along with a few false-positive lesions
	8	easily detected lesion, comparable to mild BPE on CE-MRI (main and daughter lesions)
	9	very easy identification of the lesion, resembling minimal BPE on CE-MRI
	10	very easy identification of the lesion, resembling minimal BPE on CE-MRI, with exceptionally clear main and daughter lesions

CE-MRI: contrast-enhanced magnetic resonance imaging; BPE: breast parenchymal enhancement.

**Table 2 life-13-01744-t002:** Clinical, imaging, and pathologic characteristics of total lesions (N = 186 lesions of 181 patients).

Clinical Characteristics		N (%)
Age (years)	Mean ± SD	44.6 ± 5.8
	Median (IQR)	46.0 (42.0–48.0)
MRI imaging characteristics		
Size (mm)	Mean ± SD	28.1 ± 19.3
	Median (IQR)	21.5 (15.0–36.0)
Mass/non-mass	Mass	123 (66.1)
	Non-mass	51 (27.4)
	mass and non-mass	12 (6.5)
ADC value (10^−3^ mm^2^/s)	Mean ± SD	0.9 ± 0.2
	Median (IQR)	0.9 (0.8–1.0)
CE-MRI score	Mean ± SD	5.5 ± 2.0
	Median (IQR)	6.0 (4.0–7.0)
DWI score	Mean ± SD	6.2 ± 2.3
	Median (IQR)	7.0 (4.0–8.0)
Combined score	Mean ± SD	6.6 ± 2.1
	Median (IQR)	7.0 (5.0–8.0)
Pathologic characteristics		
Pathology	DCIS	44 (23.7)
	IDC	126 (67.7)
	ILC	7 (3.8)
	Mucinous ca	6 (3.2)
	Papillary ca	1 (0.5)
	Tubular ca	2 (1.1)
Grade of invasive ca	Grade1	33 (23.2)
	Grade2	73 (51.4)
	Grade3	36 (25.4)
Grade of DCIS	Grade1	4 (9.1)
	Grade2	27 (61.4)
	Grade3	13 (29.5)
LN	Negative	133 (71.5)
	Positive	53 (28.5)
ER	Negative	26 (14.0)
	Positive	160 (86.0)
PR	Negative	36 (19.4)
	Positive	150 (80.6)
HER2	Negative	153 (82.3)
	Positive	33 (17.7)
KI-67	≤20%	108 (58.1)
	>20%	78 (41.9)
IHC type	Luminal A	93 (50.0)
	Luminal B	69 (37.1)
	Her2+	8 (4.3)
	Triple-	16 (8.6)

CE-MRI: contrast-enhanced magnetic resonance imaging; DWI: diffusion-weighted imaging; ADC: apparent diffusion coefficient; IHC: immunohistochemical; DCIS: ductal carcinoma in situ; LN: lymph node; ER: estrogen receptor; PR: progesterone receptor; HER2: human epidermal growth factor receptor 2; ca: carcinoma; SD: standard deviation; IQR: inter-quartile range.

**Table 3 life-13-01744-t003:** Detectability of main breast lesions.

	Total (N = 186)					
	MIP of CE-MRI			MIP of DWI		
	not seen (n = 21)	partially/definitely seen (n = 165)	*p* value	not seen (n = 16)	partially/definitely seen (n = 170)	*p* value
Size			0.002			0.157
Mean ± SD	18.4 ± 12.7	29.4 ± 19.6		22.4 ± 15.9	28.7 ± 19.5	
Median (IQR)	13.0 (10.0–25.0)	23.0 (16.0–37.0)		19.0 (9.5–32.0)	22.0 (15.0–36.0)	
Mass/non-mass			0.092			0.146
mass, mass and non-mass	12 (57.1)	123 (74.5)		9 (56.3)	126 (74.1)	
Non-mass	9 (42.9)	42 (25.5)		7 (43.8)	44 (25.9)	
ADC value			0.235			0.017
Mean ± SD	0.9 ± 0.2	0.9 ± 0.3		1.1 ± 0.3	0.9 ± 0.2	
Median (IQR)	0.9 (0.8–1.0)	0.9 (0.8–1.0)		1.0 (0.9–1.2)	0.9 (0.8–1.0)	
Score in CE-MRI			<0.001			<0.001
Mean ± SD	1.7 ± 0.5	5.9 ± 1.6		3.3 ± 2.0	5.7 ± 1.9	
Median (IQR)	2.0 (1.0–2.0)	6.0 (5.0–7.0)		2.5 (2.0–4.5)	6.0 (4.0–7.0)	
Score in DWI			0.002			<0.001
Mean ± SD	4.6 ± 2.7	6.5 ± 2.2		1.8 ± 0.4	6.7 ± 2.0	
Median (IQR)	4.0 (2.0–7.0)	7.0 (5.0–8.0)		2.0 (2.0–2.0)	7.0 (5.0–8.0)	
Combined score			<0.001			<0.001
Mean ± SD	4.6 ± 2.6	6.9 ± 1.8		3.3 ± 2.0	6.9 ± 1.8	
Median (IQR)	4.0 (2.0–7.0)	7.0 (6.0–8.0)		2.5 (2.0–4.5)	7.0 (6.0–8.0)	
Pathology			0.012			0.216
DCIS	10 (47.6)	34 (20.6)		6 (37.5)	38 (22.4)	
IDC + others	11 (52.4)	131 (79.4)		10 (62.5)	132 (77.6)	

CE-MRI: contrast-enhanced magnetic resonance imaging; DWI: diffusion-weighted imaging; MIP: maximal intensity projection; ADC: apparent diffusion coefficient; IDC: invasive ductal carcinoma; DCIS: ductal carcinoma in situ; Size = mm, ADC value = 10^−3^ mm^2^/s; SD standard deviation; IQR: inter-quartile range.

**Table 4 life-13-01744-t004:** Univariable and multivariable logistic regression for higher DWI scores.

	Higher DWI Scores					
	Yes (n= 85)	No (n = 101)	Univariate Odds Ratio (95% CI)	Univariate *p*	Multivariate Odds Ratio (95% CI)	Multivariate *p*
BDS						
Unmarked	72 (58.5)	51 (41.5)	reference		reference	
Marked	13 (20.6)	50 (79.4)	0.19 (0.09–0.38)	<0.001	0.18 (0.08–0.38)	<0.001
Size						
Mean ± SD	26.7 ± 16.5	29.3 ± 21.3	0.99 (0.98–1.01)	0.387		
Median (IQR)	22.0 (15.0–32.0)	21.0 (14.0–38.0)				
Size > 20 mm						
no	38 (44.2)	48 (55.8)	reference			
yes	47 (47.0)	53 (53.0)	1.12 (0.63–2.00)	0.705		
Mass/non-mass						
mass, mass and non-mass	68 (50.4)	67 (49.6)	reference		reference	
non-mass	17 (33.3)	34 (66.7)	0.50 (0.26–0.98)	0.043	0.61 (0.28–1.29)	0.196
ADC value						
Mean ± SD	0.8 ± 0.2	1.0 ± 0.3	0.08 (0.02–0.36)	0.001	0.13 (0.03–0.60)	0.009
Median (IQR)	0.8 (0.7–0.9)	0.9 (0.8–1.1)				
PATHOLOGY1				0.300		
DCIS	17 (38.6)	27 (61.4)	reference			
IDC	63 (50.0)	63 (50.0)	1.57 (0.78–3.16)	0.205		
ILC	1 (14.3)	6 (85.7)	0.36 (0.05–2.66)	0.318		
Mucinous ca	1 (16.7)	5 (83.3)	0.43 (0.06–3.30)	0.416		
Papillary ca	1 (100.0)	0 (0.0)	5.96 (0.05–768.9)	0.472		
Tubular ca	2 (100.0)	0 (0.0)	7.86 (0.18–340.1)	0.284		
PATHOLOGY2						
DCIS	17 (38.6)	27 (61.4)	reference			
IDC + others	68 (47.9)	74 (52.1)	1.44 (0.72–2.88)	0.296		
Grade in IDC (n = 142)				0.087		
Grade1	11 (33.3)	22 (66.7)	reference			
Grade2	35 (47.9)	38 (52.1)	1.80 (0.77–4.24)	0.176		
Grade3	22 (61.1)	14 (38.9)	3.04 (1.14–8.12)	0.027		
Grade in DCIS (n = 44)				0.778		
low	2 (50.0)	2 (50.0)	reference			
intermediate	11 (40.7)	16 (59.3)	0.70 (0.08–5.72)	0.737		
high	4 (30.8)	9 (69.2)	0.47 (0.05–4.63)	0.521		
LN						
Negative	58 (43.6)	75 (56.4)	reference			
Positive	27 (50.9)	26 (49.1)	1.34 (0.71–2.54)	0.370		
ER						
Negative	18 (69.2)	8 (30.8)	reference		reference	
Positive	67 (41.9)	93 (58.1)	0.33 (0.14–0.80)	0.015	0.25 (0.06–1.06)	0.060
PR						
Negative	23 (63.9)	13 (36.1)	reference		reference	
Positive	62 (41.3)	88 (58.7)	0.41 (0.19–0.86)	0.019	1.34 (0.39–4.56)	0.644
HER2						
Negative	73 (47.7)	80 (52.3)	reference			
Positive	12 (36.4)	21 (63.6)	0.64 (0.29–1.38)	0.254		
KI-67						
<20%	49 (45.4)	59 (54.6)	reference			
≥20%	36 (46.2)	42 (53.8)	1.03 (0.58–1.85)	0.915		
IHC type				0.094		
Luminal A	41 (44.1)	52 (55.9)	reference			
Luminal B	27 (39.1)	42 (60.9)	0.82 (0.43–1.54)	0.535		
Her2+	5 (62.5)	3 (37.5)	1.99 (0.45–8.72)	0.363		
Triple-	12 (75.0)	4 (25.0)	3.51 (1.08–11.48)	0.037		

DWI: Diffusion-weighted imaging; BDS: background diffusion signal; ADC: apparent diffusion coefficient; IHC: immunohistochemical; IDC: invasive ductal carcinoma; ILC: invasive lobular carcinoma; DCIS: ductal carcinoma in situ; LN: lymph node; ER: estrogen receptor; PR: progesterone receptor; HER2: human epidermal growth factor receptor 2; Size = mm; ADC value = 10^−3^ mm^2^/s.

## Data Availability

The datasets used and/or analyzed during the current study are available from the corresponding author on reasonable request.

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
