# Peer review of "The Role of Diffusion-Weighted Imaging Based on Maximum-Intensity Projection in Young Patients with Marked Background Parenchymal Enhancement on Contrast-Enhanced Breast MRI"

_life, 2023, doi:10.3390/life13081744_

Round 1

Reviewer 1 Report

The Authors investigated the role of DWI based on MIP to improve the diagnostic performance in young breast cancer patients having relevant enhancing on contrast-enhanced breast MRI. The topic is of interest, given the important role of breast MRI in the diagnostic assessment of breast lesions and as a driver of subsequent therapeutic management. I have few comments:

Introduction

1)     What do you mean by histologic enhancement ? Is it correct (line 38).

2)     I suggest to breafly explain the concept of DWI and MIP to the readers in the introduction.

3)     Why did you choose to focus on young breast cancer patients? Frequency of BPE in this population?

M&M

1)     What is your definition of young ? < 60? I am not sure it is applicable to breast cancer.

2)     I would not define as old age > 60. Please just use < 60 as definition of your population and > 60 for exclusion criteria.

3)     What do you mean by absence of surgical history? (line 63).

4)     1.303 MRI scans: But how many patients?

5)     Histopathology review (lines 147-166). Please explain all the acronyms used (ER, PR, HER2, et al)

Results

1)     Line 181-186. I would move these lines into M&M -study population section.

2)     Table 1. I assume the numbers in the third column have different dimension (years, mm, number, et al). Please specify.

3)     Table 1 and 2, captions. Please explain here also ca, SD, IQR.

Suitable for publication after revision.

Author Response

The Authors investigated the role of DWI based on MIP to improve the diagnostic performance in young breast cancer patients having relevant enhancing on contrast-enhanced breast MRI. The topic is of interest, given the important role of breast MRI in the diagnostic assessment of breast lesions and as a driver of subsequent therapeutic management. I have few comments:

Introduction

1)     What do you mean by histologic enhancement ? Is it correct (line 38).

-> We corrected it.

2)     I suggest to breafly explain the concept of DWI and MIP to the readers in the introduction.

-> As per your comment, we have provided a brief explanation of the concept of DWI and MIP in the introduction.

3)     Why did you choose to focus on young breast cancer patients? Frequency of BPE in this population?

-> As you mentioned, we chose to focus on young breast cancer patients due to the rarity of marked BPE in older patients.

M&M

1)     What is your definition of young ? < 60? I am not sure it is applicable to breast cancer.

-> I agree with your concern. However, in our institution (Korea), the frequency of breast cancer in individuals over the age of 60 is significantly lower. The elderly population (>60) represents only a small proportion, and instances of marked BPE in this age group are exceedingly rare. Therefore, for this study, we defined the elderly age group as > 60 and excluded it accordingly. We have clarified this in the abstract, Materials & Methods, and discussion sections.

2)     I would not define as old age > 60. Please just use < 60 as definition of your population and > 60 for exclusion criteria.

-> I understand your concern. In our institution (in Korea), the frequency of breast cancer among individuals over the age of 60 is significantly reduced. The elderly population (>60) comprises only a small percentage, and cases of marked BPE in this age group are rare. Therefore, for this study, we have defined the old age group as > 60 and excluded it from our analysis. We have provided a clarification in the abstract, Materials & Methods, and discussion sections.

3)     What do you mean by absence of surgical history? (line 63).

-> It refers to the absence of surgical confirmation for the diagnosed cases. As you pointed out, we have now clarified this in the manuscript.

4)     1.303 MRI scans: But how many patients?

-> It represents the same number of 1,303 MRI scans performed on 1,303 individual patients.

5)     Histopathology review (lines 147-166). Please explain all the acronyms used (ER, PR, HER2, et al)

-> As per your comment, we have now provided explanations for all the acronyms used (ER, PR, HER2, et al).

Results

1)     Line 181-186. I would move these lines into M&M -study population section.

-> As you suggested, we have relocated these lines into the Materials & Methods section.

2)     Table 1. I assume the numbers in the third column have different dimension (years, mm, number, et al). Please specify.

-> As per your comment, we have now specified the dimensions for each number in the third column of Table 1.

3)     Table 1 and 2, captions. Please explain here also ca, SD, IQR.

-> As you pointed out, we have provided explanations for the abbreviations "ca" (carcinoma), "SD" (standard deviation), and "IQR" (interquartile range) in the captions of both Table 1 and Table 2.

Reviewer 2 Report

Dear Authors

The article presents a very important problem as well as routine MRI examinations. However, I am missing some information in the description, please complete:

In the introduction: the cited articles do not reflect the theory of MRI DW, whether it differs from other techniques and what it consists of.

I suggest adding what the sequence looks like.

In the materials and method, the description of the measurement parameters must be corrected to descriptive and the units should be next to each value

Please complete the conclusions. Thank you.

Author Response

Dear Authors

The article presents a very important problem as well as routine MRI examinations. However, I am missing some information in the description, please complete:

In the introduction: the cited articles do not reflect the theory of MRI DW, whether it differs from other techniques and what it consists of.

I suggest adding what the sequence looks like.

-> Thank you for your feedback. I have made sure to address your suggestions in the revised manuscript, covering the introduction, Materials & Methods, and discussion sections.

In the materials and method, the description of the measurement parameters must be corrected to descriptive and the units should be next to each value

-> Thank you for your valuable input. I have made sure to address your comments and ensure the accuracy and clarity of the measurement parameters and units in the revised manuscript.

Please complete the conclusions. Thank you.

-> As per your comment, we completed the conclusions.

Reviewer 3 Report

“The Role of Diffusion-Weighted Imaging Based on Maximum Intensity Projection in Young Patients with Marked Background Parenchymal Enhancement on Contrast-Enhanced Breast MRI”

The manuscript presented for review consists of 14 pages with 30 references. 3 tables and 6 figures are included. The study is original. The manuscript is divided into 5 sections. The work fits the journal scope, however requires major improvement. 

Abstract: A background of the study is not described.

Line 120-135: I would recommand to present the scale in the table. 

- Line: 192-194 - text inserted by a mistake?

-Figure 1 - poorly readable - I would recommand to increase a font. 

-Figure 2 - numbering photos from top to bottom and marking the order in the legend could make more clear the presented results.

-Table 1,2,3 - separating results with horizontal lines could make more clear the presented results. 

-References: Are there more recent studies concerning the subject?

- In abstract: past tense should be used.

- In abstract: ''partially or completely seen" - repetition.

- Line 46: 'reducing false positives"... results?

- Line 49-56: ''is becoming increasingly popular'' -> has becoming?

I would suggest to check the article by a native speaker.

Author Response

“The Role of Diffusion-Weighted Imaging Based on Maximum Intensity Projection in Young Patients with Marked Background Parenchymal Enhancement on Contrast-Enhanced Breast MRI”

The manuscript presented for review consists of 14 pages with 30 references. 3 tables and 6 figures are included. The study is original. The manuscript is divided into 5 sections. The work fits the journal scope, however requires major improvement.

-Abstract: A background of the study is not described.

-> As you suggested, we added a background in the abstract.

-Line 120-135: I would recommand to present the scale in the table.

-> As you pointed out, we have now presented the scale in the table, as per your comment.

-Line: 192-194 - text inserted by a mistake?

-> As you correctly noted, those lines were inserted by mistake, and we have now deleted them from the manuscript.

-Figure 1 - poorly readable - I would recommand to increase a font.

-> As you pointed out, we have now increased the font size in Figure 1, as per your comment.

-Figure 2 - numbering photos from top to bottom and marking the order in the legend could make more clear the presented results.

-> As you suggested, we have now numbered the photos in Figure 2 from top to bottom and marked the order in the legend to improve clarity.

-Table 1,2,3 - separating results with horizontal lines could make more clear the presented results.

-> As per your comment, we have now added horizontal lines to separate the results in Table 1, 2, and 3 to improve clarity.

-References: Are there more recent studies concerning the subject?

-> We thoroughly searched for more recent references related to the subject, but the most recent one we found was already included in our manuscript. The reference is as follows; Background parenchymal enhancement at postoperative surveillance breast MRI: association with future second breast cancer risk S. H. Lee, M.-j. Jang, H. Yoen, Y. Lee, Y. S. Kim, A. R. Park, et al. Radiology 2023 Vol. 306 Issue 1 Pages 90-99

Comments on the Quality of English Language

- In abstract: past tense should be used.

-> In the abstract, we have made the necessary edits to use past tense, as you pointed out.

- In abstract: ''partially or completely seen" - repetition.

-> In the abstract, we have simplified the phrase "partially or completely seen" to avoid repetition, as you pointed out.

- Line 46: 'reducing false positives"... results?

-> As you suggested, we have now deleted the phrase "reducing false positives" from the manuscript.

- Line 49-56: ''is becoming increasingly popular'' -> has becoming?

-> As you pointed out, we have made the necessary change to "has become increasingly popular" to ensure the correct tense.

I would suggest to check the article by a native speaker.

-> As per your recommendation, we have had the article checked by a native English speaker, and we have also obtained a 'certificate of editing' to ensure the quality and accuracy of the language.

Reviewer 4 Report

The purpose of this study is to investigate whether diffusion-weighted imaging  utilizing maximum intensity projection (MIP) plays a supportive role in young patients with marked background parenchymal enhancement (BPE) on contrast-enhanced MRI (CE-MRI). The paper is clearly presented and justified. Only the language must be revised for the acceptance.

The manuscript must be proofread.

Author Response

The purpose of this study is to investigate whether diffusion-weighted imaging utilizing maximum intensity projection (MIP) plays a supportive role in young patients with marked background parenchymal enhancement (BPE) on contrast-enhanced MRI (CE-MRI). The paper is clearly presented and justified. Only the language must be revised for the acceptance.

-> Thank you for your review. We are grateful for your feedback, and we agree that the paper is well-presented and justified. As per your suggestion, we have carefully revised the language to ensure its acceptance. We have also taken the extra step of having the English checked by a native speaker and obtained a 'certificate of editing' to further validate the quality of the language.

Round 2

Reviewer 1 Report

I do not have any further comment

Moderatwe english revision required

Reviewer 2 Report

Thank you

Reviewer 3 Report

The Authors have included all suggestions. 

In my opinion the study in a present form may be acceptem for publication.

I appreciate Authors effort in improving the quality of an article.